

# Comparison of ultrafiltration and iron chloride flocculation in the preparation of aquatic viromes from contrasting sample types

Kathryn Langenfeld[1], Kaitlyn Chin[1], Ariel Roy[1], Krista Wigginton[1] and Melissa B. Duhaime[2]

[1] Department of Civil and Environmental Engineering, University of Michigan - Ann Arbor, Ann Arbor, MI, United States of America
[2] Department of Ecology and Evolutionary Biology, University of Michigan - Ann Arbor, Ann Arbor, MI, United States of America

## ABSTRACT

Viral metagenomes (viromes) are a valuable untargeted tool for studying viral diversity and the central roles viruses play in host disease, ecology, and evolution. Establishing effective methods to concentrate and purify viral genomes prior to sequencing is essential for high quality viromes. Using virus spike-and-recovery experiments, we stepwise compared two common approaches for virus concentration, ultrafiltration and iron chloride flocculation, across diverse matrices: wastewater influent, wastewater secondary effluent, river water, and seawater. Viral DNA was purified by removing cellular DNA via chloroform cell lysis, filtration, and enzymatic degradation of extra-viral DNA. We found that viral genomes were concentrated 1-2 orders of magnitude more with ultrafiltration than iron chloride flocculation for all matrices and resulted in higher quality DNA suitable for amplification-free and long-read sequencing. Given its widespread use and utility as an inexpensive field method for virome sampling, we nonetheless sought to optimize iron flocculation. We found viruses were best concentrated in seawater with five-fold higher iron concentrations than the standard used, inhibition of DNase activity reduced purification effectiveness, and five-fold more iron was needed to flocculate viruses from freshwater than seawater—critical knowledge for those seeking to apply this broadly used method to freshwater virome samples. Overall, our results demonstrated that ultrafiltration and purification performed better than iron chloride flocculation and purification in the tested matrices. Given that the method performance depended on the solids content and salinity of the samples, we suggest spike-and-recovery experiments be applied when concentrating and purifying sample types that diverge from those tested here.

# INTRODUCTION

Viruses are important members of natural and engineered aquatic ecosystems that can outnumber other microbes by up to two orders of magnitude (*Parsons et al., 2012*;

Corresponding authors
Krista Wigginton, kwigg@umich.edu
Melissa B. Duhaime, duhaimem@umich.edu

*Wigington et al., 2016*; *Cael et al., 2018*), and influence their host's ecology and evolution through metabolic reprogramming and mortality (*Breitbart, 2012*; *Abedon, 2008*). To better understand the fate and role of viruses in aquatic systems, whole community sequencing ('metagenomics') is used for the untargeted exploration of viruses in a community context (*Dion, Oechslin & Moineau, 2020*). Metagenomics has led to the unprecedented discovery of viral diversity and function (*Brum et al., 2015*; *Roux et al., 2016b*; *Koonin & Yutin, 2020*; *Dutilh et al., 2014*), but its ability to deliver an unbiased representation of viral communities is nonetheless hindered by methodological challenges and biases (*Duhaime & Sullivan, 2012*).

The central challenge is that viral DNA comprises a small fraction of total community DNA due to viral genomes being one to four orders of magnitude smaller than eukaryotic and prokaryotic genomes (*Mahmoudabadi & Phillips, 2018*). As a result, in the preparation of virus-enriched metagenomes, 'viromes', it can be difficult to recover sufficient viral genomic material to generate sequencing libraries without biases introduced by amplification steps (*Roux et al., 2016b*; *Brinkman et al., 2018*). Some studies avoid this obstacle by studying viruses in metagenomes prepared from whole community DNA, rather than viromes. But this approach may not be appropriate for all research questions, as it will capture temperate and actively replicating viruses, rather than predominantly free viruses. Further, when viral genomes are sequenced amidst the overwhelming cellular DNA, the sequencing effort dedicated to viruses is drastically limited. Purifying samples to increase the ratio of viral DNA to cellular DNA results in a more comprehensive representation of the viral community. This enhancement will increase the likelihood of sequencing low abundance and rare viruses and increase the sensitivity of viral detection studies. Overall, several of these challenges can be mitigated during sample preparation by efficiently concentrating and purifying viral genomes to focus maximal sequencing effort on viruses.

A number of methods to concentrate and purify viruses have been developed and are broadly used. Viruses are concentrated in water by exploiting their unique physical and structural properties, such as size (20 to 300 nm) and surface charge (commonly negative), with varying degrees of effectiveness. Methods that rely on charge include flocculation or precipitation (iron chloride, skimmed milk, lanthanum chloride, aluminum sulfate, aluminum chloride, and polyethylene glycol) (*John et al., 2011*; *Hjelmso et al., 2017*; *Calgua et al., 2013*; *Zhang et al., 2013*; *Ye et al., 2016*; *Randazzo et al., 2019*) and virus adsorption-elution (*Hjelmso et al., 2017*; *Kunze et al., 2015*; *Shi et al., 2016*; *Millen et al., 2012*). Ultrafiltration takes advantage of particle sizes to concentrate viruses in dead-end, tangential, or axial flow configurations (*Calgua et al., 2013*; *Ye et al., 2016*; *Kunze et al., 2015*; *Shi et al., 2016*; *Hill et al., 2005*; *Rhodes et al., 2016*; *Hurwitz et al., 2013*; *Pei et al., 2012*; *Thurber et al., 2009*; *Smith & Hill, 2009*; *Gallardo, Morris & Rhodes, 2019*). Alternate virus concentration methods that do not rely on size or surface charge include ultracentrifugation and lyophilization (*Hjelmso et al., 2017*; *Calgua et al., 2013*; *Ye et al., 2016*). Most of these concentration methods were developed for PCR-based detection where targets are selectively amplified and thus purification is not necessary. When purification is conducted on virome samples, non-viral biological material is removed

from the aqueous samples via a number of approaches. For instance, submicron filtration can be used to separate cells from viruses, chloroform can be used to solubilize lipids in the cell membrane and cause cell lysis (*Breitbart & Rohwer, 2005*; *Hannigan et al., 2018*; *Kauffman et al., 2018*), non-encapsidated extra-viral DNA can be enzymatically degraded (*Maruyama, Oda & Higashihara, 1993*), and density gradients can be used to separate phages from cells based on their buoyant densities (*Hurwitz et al., 2013*; *Kleiner, Hooper & Duerkop, 2015*; *Trubl et al., 2019*).

Despite decades of research on virus concentration and purification methods, a knowledge gap remains regarding the preparation of virome samples. Of the existing studies that have evaluated aquatic virome sample preparation (*Hjelmso et al., 2017*; *Hurwitz et al., 2013*; *Uyaguari-Diaz et al., 2016*), few have applied comparable methods or assessed performance at each of the concentration and purification steps, making cross-study comparisons difficult. While a multitude of studies have demonstrated the ability to produce aquatic viral metagenomes using various protocols (*Hjelmso et al., 2017*; *Hurwitz et al., 2013*; *Uyaguari-Diaz et al., 2016*; *Wang et al., 2020*), none have evaluated viral recovery or removal of cellular DNA. Typically, these method assessments have focused on a single aquatic matrix. The performance, and thus suitability, of different concentration and purification methods across a variety of water sample types is limited.

In this study, we evaluated the impact of sample matrix on the performance of two commonly applied concentration methods, ultrafiltration and iron chloride flocculation. Ultrafiltration and iron chloride flocculation were selected because they have been widely used for marine and freshwater virome studies (*Brum et al., 2015*; *Hurwitz et al., 2013*; *Beaulaurier et al., 2020*; *Gregory et al., 2019*; *Moon et al., 2020a*; *Moon et al., 2020b*; *Aguirre de Carcer et al., 2015*). Using virus spike-and-recovery experiments, a step-wise assessment of each method was performed for four contrasting sample types that varied in their solids content and salinity: wastewater influent (i.e., raw sewage), wastewater secondary effluent (i.e., post carbon removal, pre-disinfection), river water, and seawater. Our findings will inform the sample preparation of future virome studies, especially as the quantitative rigor of metagenomics is further pursued.

## MATERIALS & METHODS

### Sample collection

Grab samples of secondary effluent and raw influent were collected from automatic samplers at the Ann Arbor wastewater treatment plant (Ann Arbor, MI) in November 2018 through November 2019 (Table S1). Grab samples of river water were collected from a boat ramp upstream of the Ann Arbor wastewater discharge site along the Huron River in Ann Arbor at the surface from May through July 2019 (Table S1). Raw influent, secondary effluent, and river water were collected in autoclaved bottles and carboys. Samples were transported to the laboratory on ice within 1 h of collection and began processing immediately upon arrival in the lab. Seawater was collected from the Shedd Aquarium (Chicago, IL) on February 27, 2020 and immediately transported to the lab in Ann Arbor, MI on ice. Samples were stored at 4 °C until processing for a maximum of one week (Table S2).

## Sample characterization

Sample volumes were determined in the lab by weighing the sample. Immediately prior to processing, the pH was measured with a Mettler Toledo pH meter calibrated with a 4, 7, and 10 pH standards prior to measurement (Table S1). The total suspended solids (TSS) and volatile suspended solids (VSS) were determined for each sample using standard methods with 20 mL of influent or river water in duplicate and 40 mL of secondary effluent stored at −20 °C until analysis (Table S1) (*Rice, Baird & Eaton, 2017*). The seawater was tested for changes in pH during the week of storage (Table S2).

## Phages for spike and recovery

Phage spike-and-recovery experiments evaluated the performance of concentration and purification methods. Several different phages were spiked into the matrices to estimate the total viral recovery: *Enterobacteria* phage T3 (GenBank accession no. NC_003298, ATCC® BAA-1025-B1^TM), *Enterobacteria* phage T4 (GenBank accession no. NC_000866), *Enterobacteria* phage PhiX174 (GenBank accession no. NC_001422), *Pseudoalteromonas* phage HS2 (GenBank accession no. KF302036), *Pseudoalteromonas* phage HM1 (GenBank accession no. KF302034.1), and *Sulfitobacter* phage ICBM5, a ssDNA *Microviridae* phage provided by the Moraru Phage Lab (Institute for Chemistry and Biology of the Marine Environment, Oldenburg, Germany) (Table 1). Phages originating from freshwater (T3, T4, and PhiX174) and seawater (HS2, HM1, and ICBM5) were spiked into freshwater and seawater, respectively (Table 2), to avoid compromising the integrity of protein capsids due to salinity differences, as was observed in preliminary experiments (SI section S3). For secondary effluent and seawater samples, multiple phages were spiked into the same samples. T3, T4, HS2, and HM1 have tails protruding from the protein capsid (*Duhaime et al., 2017*; *Matsuo-Kato, Fujisawa & Minagawa, 1981*; *Leiman et al., 2004*) and PhiX174 and ICBM5 have small icosahedral capsids (*Dion, Oechslin & Moineau, 2020*). None of the phages were enveloped.

All phages were cultured prior to spike-and-recovery studies. T3 and T4 host (*E. coli* ATCC® 11303) and PhiX174 host (*E. coli* ATCC® 13609) were grown overnight in 25 mL of host media (Table S3) at 37 °C and 180 rpm from glycerol stocks. HS2 host (*Pseudoalteromonas* sp. 13-15), HM1 host (*Pseudoalteromonas* sp. H71), and ICBM5 host (*Sulfitobacter* sp. SH24-1b) were suspended in 25 mL of host media (Table S3) from culture plates and allowed to grow overnight at 25 °C and 180 rpm. The plaque overlay method generated plates of completely lysed bacterial lawns for each phage. Briefly, 100 μL of $10^6$ pfu mL$^{-1}$ T3, T4, or PhiX174 were combined with 100 μL of respective host in 3.5 mL of soft nutrient agar and poured over a hard nutrient agar plate (Table S3), then incubated at 37 °C overnight. For HS2, HM1, and ICBM5, the same plaque overlay method was used with 300 μL of respective host and incubation at 25 °C overnight. The top layer of soft agar was gently collected and 5 mL of respective buffer (Table S3) was poured on each completely lysed plate. The plates were gently mixed and incubated on the benchtop for 20 min. The buffer was combined with the soft agar, gently shaken, and incubated at 4 °C for 2 h. The mixture was vortexed for 30 s and treated with 0.5 mL chloroform, vigorously mixed for 2 min, and centrifuged (Table S4). The supernatant was collected and aerated to
**Table 1** **Three freshwater phages and three marine phages were used in spike-and-recovery experiments.** Characteristics of phages spiked into water samples to access viral recovery during concentration and purification processes.

| Phage | Genome length (bp) | Genome composition | Length (nm) |
|---|---|---|---|
| *Enterobacteria* phage T3 | 38,208 | dsDNA | 70 (head and tail) |
| *Enterobacteria* phage T4 | 168,903 | dsDNA | 203 (head and tail) |
| *Enterobacteria* phage PhiX174 | 5,386 | ssDNA | 26 |
| *Pseudoalteromonas* phage HS2 | 38,208 | dsDNA | 210 (head and tail) |
| *Pseudoalteromonas* phage HM1 | 129,401 | dsDNA | ∼200 (head and tail) |
| *Sulfitobacter* phage ICBM5 | 5,581 | ssDNA | 29 |

**Table 2** **Phage spike additions for concentration and purification method evaluation.** See Table 1 for characteristics.

| Matrix | Phage spiked | |
|---|---|---|
| | Iron chloride flocculation and purification | Ultrafiltration and purification |
| Influent | T3 | T3 |
| Secondary Effluent | T3 | T3, T4, PhiX174 |
| River Water | T3 | T3 |
| Seawater | HS2, HM1, ICBM5 | HS2, HM1, ICBM5 |

remove trace chloroform. Stocks were 0.45-$\mu$m (T3, T4, HM1, HS2, ICBM5) or 0.22-$\mu$m (PhiX174) polyethersulfone (PES) filtered (CellTreat Scientific Products, Cat. No. 229771 and 229747, respectively) and stored at 4 °C.

## Optimizing iron chloride flocculation

To optimize iron chloride flocculation, modified jar tests were performed in triplicate for each sample matrix. Samples were pre-filtered through a 100-$\mu$m Long-Life filter bag for water made of polyester felt (McMaster-Carr, Cat. No. 6835K58) then 0.45-$\mu$m Express PLUS PES filters (MilliporeSigma$^{TM}$, Cat. No. HPWP09050 or HPWP14250) and aliquoted in 500 mL increments into 6 autoclaved glass bottles with stir bars. T3 and HS2 were spiked into each jar at approximate concentrations of $5 \times 10^5$ gene copies (gc) $\mu L^{-1}$ for freshwater and $10^7$ gc $\mu L^{-1}$ for seawater, respectively. Samples were collected immediately following the spike addition for recovery analysis. A sterile 10 g Fe $L^{-1}$ iron chloride solution was made immediately prior to use. For consistency with the previously established method, half of the iron dose was added to each jar followed by a minute of turbulent mixing on a stir plate and then repeated (*John, Poulos & Schirmer, 2015b*). Iron concentrations of 0, 0.1, 1, 5, 10, and 25 mg Fe $L^{-1}$ were tested (Table S5). Following the addition of iron, samples were left for an hour at room temperature, then flocs were captured on 0.45-$\mu$m Express PLUS filters. Samples of the filtrate, or material passing through the filter, were collected for recovery analysis (Fig. 1). The floc-containing filters were carefully placed in sterile centrifuge tubes along with freshly-made 1x oxalic acid resuspension buffer, as described previously (*John, Poulos & Schirmer, 2015a*) (Table S5). All samples were placed
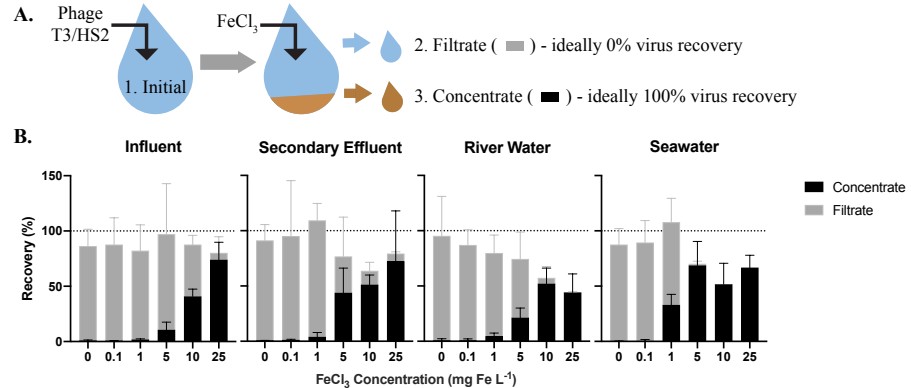

**Figure 1** **The efficiency of virus flocculation was tested with jar tests at several iron chloride concentrations.** (A) T3 for freshwater matrices and HS2 for seawater was spiked into jars and an initial sample was collected to determine the initial T3 or HS2 genome concentration (1). Iron chloride was added to the samples, which underwent turbulent mixing, then flocs were captured on filters. A sample of the water flowing through the filter (the 'filtrate' (2)) was collected to determine the number of T3 or HS2 genomes not flocculated. The filter was placed in resuspension buffer to dissolve flocs. A concentrate sample (3) was then collected to determine the number T3 or HS2 captured in flocs and successfully resuspended. (B) The recoveries of T3 and HS2 genomes in the sample concentrate (black) and filtrate (gray). Bars are stacked. Error bars represent the standard deviations around the geometric mean of experimental triplicates. A perfect recovery of T3 or HS2 genomes would result in the filtrate and concentrate bars stacking to the 100% recovery dashed line.

on a shaker table in the dark at 4 °C and 180 rpm overnight to dissolve the flocs. The concentrate (i.e., the resuspension buffer with dissolved flocs), was separated from the filters as described previously (*John, Poulos & Schirmer, 2015b*) and samples were collected for recovery analysis (Fig. 1).

## Iron chloride flocculation and purification method (Fig. 2A)
### Step 1: iron chloride flocculation
The protocol was adapted from *John et al. (2011)* to concentrate viruses. 500-mL influent, secondary effluent, and river water samples and 20-L seawater samples were pre-filtered through 0.45-μm Express PLUS filters. Rather than 0.22-μm pore size filters, 0.45-μm pore size filters were used in this step to recover giant phages (*Schultz et al., 2017*; *Uchiyama et al., 2014*). Half of the iron required for the best concentration, as determined in the jar tests (above), was added to the sample. Samples were rapidly mixed for a minute with a magnetic stir bar for freshwater matrices or shaken in carboys for seawater. This process was repeated with the remaining half of the required iron. Samples were left at room temperature for 1 h to flocculate. The flocculated viruses were captured on 0.45-μm Express PLUS PES filters. PES matrix filters were used for this step because of their superior flow rates, as compared to the limited flow rates achieved with PC isopore filters, an observation consistent with prior knowledge (*Ho & Zydney, 1999*). Further, preliminary experiments comparing 0.22-μm PES filters versus 1-μm polycarbonate filters demonstrated the PES filter had superior flow rates (2.9-fold greater than the PC) and equivalent recoveries of nucleic acids, as measured by a Nanodrop spectrophotometer (data not included). The

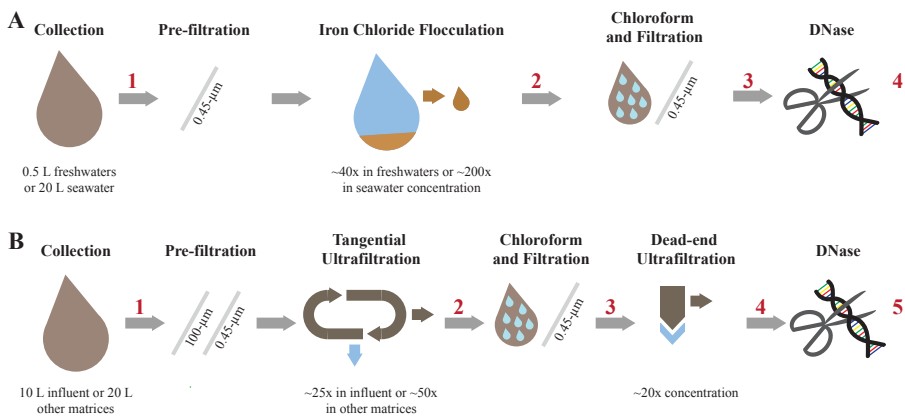

**Figure 2** **Overview of each step involved in concentrating and purifying viruses with iron chloride flocculation and ultrafiltration.** Concentration and purification process for the iron chloride flocculation and purification method (A) and the ultrafiltration and purification method (B). The red numbers indicate where aliquots were collected to measure viral genome and 16S rRNA gene copy concentrations.

filters were carefully folded into sterile centrifuge tubes. 1 mL of 1x oxalic acid resuspension buffer per mg Fe used to flocculate the sample was added to the filter containing tubes. The viruses were resuspended from the filters by shaking at 4 °C and 180 rpm overnight. The solution was transferred to a clean centrifuge tube as described previously (*John, Poulos & Schirmer, 2015b*). Only 500-mL was used for freshwater samples because the concentration by volume is independent of initial sample volume (i.e., mass of iron added controls the volume of the resuspension buffer added) and a final volume of 12.5 mL is sufficient for subsequent steps.

### Step 2: Chloroform and filtration

To isolate the viruses in the sample, cells were lysed with chloroform. 1 mL chloroform was added to the sample and vortexed for approximately 2 min. Chloroform settled out of suspension by sitting undisturbed on the benchtop for 15 min. The bulk of the chloroform was pipetted off the bottom of the sample and disposed. The trace remainder of chloroform was aerated from the sample in a fume hood for approximately 10 min. The cell debris was filtered from the sample with 0.45-μm Express PLUS filters.

### Step 3: DNase treatment

DNase treatment was performed with a previously established method with the reaction time reduced to one hour (*Thornton, 2015*). Briefly, lyophilized DNase 1, grade II from bovine pancreas (Roche, Cat. No. 10104159001) was resuspended in a storage buffer (10mM Tris-Cl pH 7.5, 2 mM $CaCl_2$ in 50% glycerol) to a concentration of 40,000 U mL$^{-1}$ and stored at −20 °C. Immediately prior to DNase treatment of sample, DNase 1 in storage buffer was diluted 1:40 in a 10x reaction buffer (100 mM Tris–HCl pH 7.6, 25 mM $MgCl_2$, 5 mM $CaCl_2$) and gently mixed. 100 U mL$^{-1}$DNase 1 in the reaction buffer was added to the sample and reacted for 1 h on the bench top. The DNase reaction was inhibited with the addition of 100 mM EDTA and 100 mM EGTA. Chloroform and DNase treatments

were performed to purify the viral nucleic acids as tested in a purification optimization experiment provided in the SI section 4.

## Ultrafiltration and purification method (Fig. 2B)
### Step 1: Tangential ultrafiltration
10 L influent or 20 L of other matrices were pre-filtered through 100-$\mu$m polyester filter bag then filtered through 0.45-$\mu$m Express PLUS filters to remove large particles. Tangential ultrafiltration was performed with hollow-fiber ultrafilters, specifically Dialyzer Rexeed single use dialysis filters with a surface area of 2.5 m$^2$ and approximate molecular weight cut off of 30 kDa (Asahi Kosei Medical Co., Ltd, Cat. No. 6292966), as described previously (*Hill et al., 2005*). Briefly, the sample and filter were configured such that the sample passed through the filter tangential to the membrane surface. The sample volumes progressively decreased as water passed through the membrane pores and particles were retained. New filters were used for each sample. The sample flowed through the filter in the direction labeled blood until the sample volume was minimized and air began to enter the tubing. The minimized volume was approximately 350 mL. At the minimal volume, the flow direction was reversed, and the concentrated sample was collected. The exact volume after tangential ultrafiltration was determined by weighing the sample and assuming a density of 1 g mL$^{-1}$.

### Step 2: Chloroform and filtration
The same chloroform and filtration method as performed during the iron chloride flocculation and purification method was implemented in this method.

### Step 3: Dead-end ultrafiltration
The sample was concentrated an additional 20-fold with dead-end ultrafiltration. Dead-end ultrafiltration was performed with 100 kDa MWCO and 1 cm$^2$ surface area Amicon$^{TM}$ Ultra Centrifugal filter units (MilliporeSigma$^{TM}$, Cat. No. UFC510096). Four new filters were used for each sample and processed in parallel. The sample was centrifuged at 3,000*xg* and 4 °C while incrementally refilling the filters until 4 mL of sample was reduced to 200 $\mu$L per filter. Additional pre-washing, incubation, or sonication steps were not included in the protocol because results from preliminary experiments indicated that pre-washing the filter with water, incubating concentrate with BSA, and sonicating the filter cartridge prior to collecting concentrate did not improve viral genome recoveries (SI section 5; Fig. S2). The concentrate was collected by inverting the filter into a clean collection tube and centrifuging at 1,000*xg* for 1 min. The contents from individual filters were combined. No additional treatment of the ultrafilters was performed prior or following dead-end ultrafiltration in accordance with the results from a dead-end ultrafiltration optimization experiment summarized in SI section 5.

### Step 4: DNase treatment
The same DNase treatment as performed during the iron chloride flocculation and purification method was implemented for this method.

## Phage concentration and 16S rRNA removal analysis

Phages were spiked into samples for an approximate concentration of $10^4$ gc $\mu L^{-1}$ prior to pre-filtering to monitor the recovery after each step of iron chloride flocculation and purification and ultrafiltration and purification (Table 2). We used removal of the ~500 bp long 16S rRNA V3 region amplicon to approximate the removal of non-viral DNA in the sample. 16S rRNA is commonly used to estimate total bacteria counts in samples because it is a conserved region of bacterial genomes (*Nadkarni et al., 2002*). A sample was collected prior to the phage addition to examine background concentrations of spiked phages in the samples. An "initial" sample was collected immediately after spike additions to determine the exact concentration of each phage representing total phage recovery. For the iron chloride flocculation and purification method, additional samples were collected after 0.45-μm filtering and iron chloride flocculation, chloroform and 0.45-μm filtering, and DNase treatment (Fig. 3A). Samples were collected after 0.45-μm filtering and tangential ultrafiltration, chloroform and 0.45-μm filtering, dead-end ultrafiltration, and DNase treatment for the ultrafiltration and purification method (Fig. 3B). The initial 16S rRNA concentration was determined with the "initial" sample for iron chloride flocculation and purification and ultrafiltration and purification experiments. Four key parameters were calculated to assess concentration and purification performance: virus concentration factor, virus recovery, 16S rRNA concentration factor, and virus to 16S rRNA enrichment. The virus concentration factor is the concentration of viral genomes after a step divided by the initial concentration (Eq. (1)). The virus recovery builds from the virus concentration factor by accounting for the change in volume occurring throughout the concentration and purification processes to identify losses of virus (Eq. (2)). The same virus concentration factor calculation was applied to 16S rRNA gene copies to calculate the 16S rRNA concentration factor (Eq. (3)). Lastly, the virus to 16S rRNA enrichment is the ratio of the virus concentration factor and the 16S rRNA concentration factor to determine if viral genomes were selectively concentrated throughout the processes (Eq. (4)). Gene copy concentrations were determined with ddPCR probe assays for iron chloride flocculation and purification and ultrafiltration and purification methods or qPCR SYBR green assays for optimizing iron chloride flocculation defined in *T3 and HS2 qPCR assays* and *Phage and 16S rRNA ddPCR assays* sections.

$$Virus\ Concentration\ Factor = \frac{[Phage]_{Step}\left(gc\ \mu L^{-1}\right)}{[Phage]_{Initial}\left(gc\ \mu L^{-1}\right)} \tag{1}$$

$$Recovery\ (\%) = Virus\ Concentration\ Factor \cdot \frac{Volume_{Step}\left(mL\right)}{Volume_{Initial}\left(mL\right)} \cdot 100\% \tag{2}$$

$$16S\ rRNA\ Concentration\ Factor = \frac{[16S\ rRNA]_{Step}\left(gc\ \mu L^{-1}\right)}{[16S\ rRNA]_{Initial}\left(gc\ \mu L^{-1}\right)} \tag{3}$$

$$Virus\ to\ 16S\ rRNA\ Enrichment = \frac{Virus\ Concentration\ Factor}{16S\ rRNA\ Concentration\ Factor} \tag{4}$$

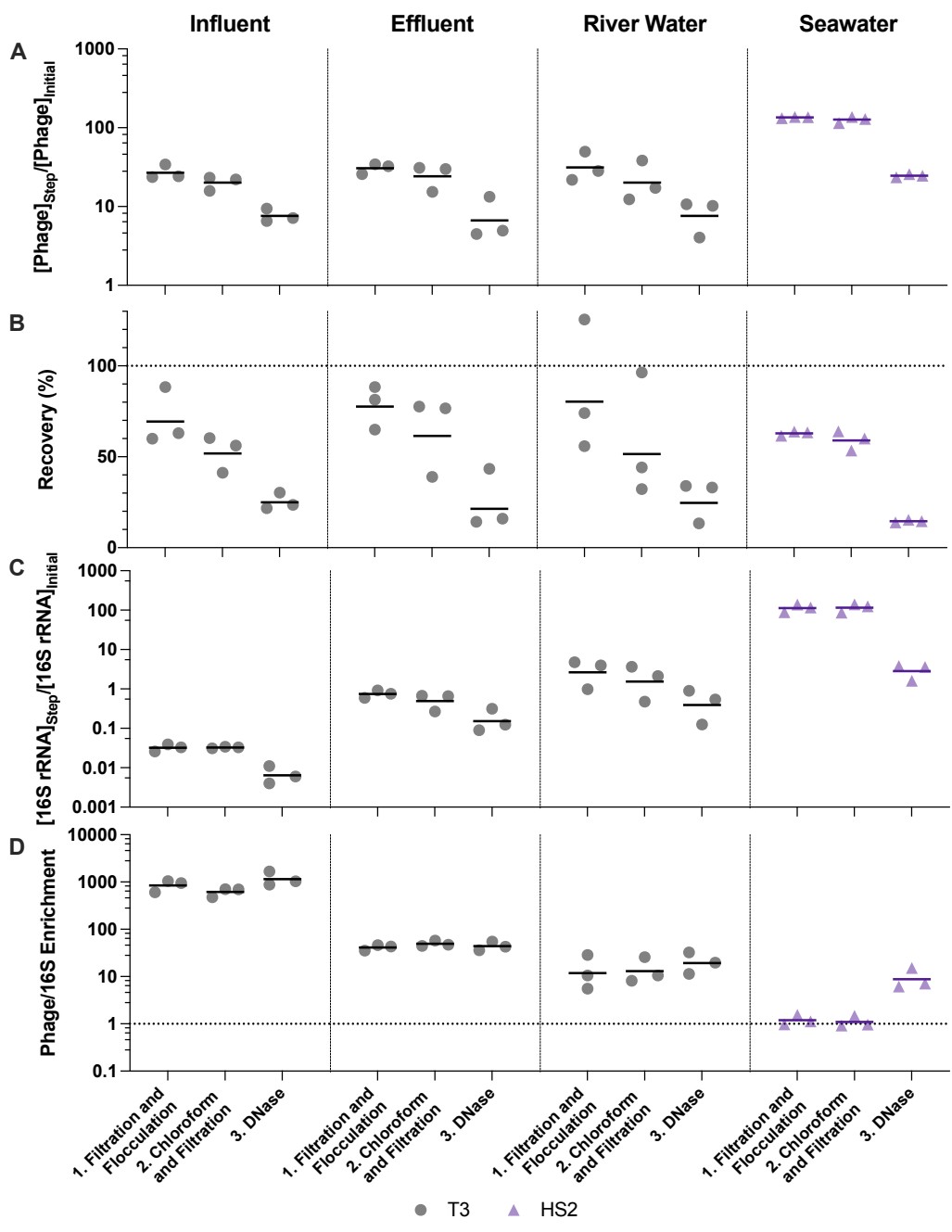

**Figure 3 Stepwise evaluation of iron chloride flocculation and purification with 40-fold concentration of freshwater matrices and 200-fold concentration of seawater.** 25 mg Fe L$^{-1}$ was used for freshwater matrices and 5 mg Fe L$^{-1}$ was used for seawater during iron chloride flocculation. (A) T3 or HS2 concentration factors at each step on a log-scale. Values greater than 1 indicated an increase in viral genomes. (B) T3 and HS2 recovery for each step of iron chloride flocculation demonstrated step-wise viral loss. Perfect recovery is indicated by the dotted line at 100%. (C) 16S rRNA concentration factor on a log-scale indicated non-viral DNA removal. Concentration factor values less than 1 indicated reduction of 16S rRNA gene copies. (D) T3 or HS2 to 16S rRNA enrichment with each step on a log-scale, as calculated by the virus concentration factor divided by the 16S rRNA concentration factor. Enrichments greater than 1 demonstrated viral genomes were concentrated more than 16S rRNA gene copies during the process. Individual measurements are points with a bar indicating the geometric mean of experimental replicates.

## DNA extraction

DNA extraction was performed with QIAamp UltraSens Virus Kit (QIAGEN, Cat. No. 53706). The manufacturer's protocol was followed with minor changes. The first six steps were modified to combine 140 μL of sample with 5.6 μL carrier RNA and vortexed briefly. All DNA extractions occurred within 3 h of sample generation.

## T3 and HS2 qPCR assays

Primers (5′ to 3′) specific to T3 were selected (351 bp; forward, CCA ACG AGG GTA AAG TGA TAG; reverse, CGA CGA TAG CGA ATA GGA TAA G). Primers specific to HS2 were selected (300 bp; forward, GGT TGA TGA AAA GTC ACT; reverse, CGG GGC AGA TCT AAA TGA). The 10 μL reaction contained 5 μL 2x Biotium Fast-Plus EvaGreen master mix, 0.5 μM T3 or HS2 primers, 0.625 mg mL$^{-1}$ bovine serum albumin, and 1 μL of DNA template. Standard curves were prepared in triplicate between 100 and 10$^6$ gene copies μL$^{-1}$ with gBlocks dsDNA fragments of the amplicon sequence (IDT, Coralville, IA) (Table S7). Ten replicates of the previously determined limit of quantification (T3 = 100 gene copies μL$^{-1}$, HS2 = 30 gene copies μL$^{-1}$) were measured on each plate, and two ddH$_2$O negative controls and two positive controls of DNA extracts from virus stocks were included on each plate. All positive controls were positive and all negative controls were negative for T3 or HS2. qPCR was performed with the realplex$^2$ Mastercycler epgradient S automated real-time PCR system (Eppendorf®, New York City, NY) with standard reaction conditions (Table S8). All efficiencies were greater than 80% and R$^2$ values were greater than 0.98. Inhibition was assessed by comparing measurements for a random sampling of undiluted and 1:10 diluted freshwater DNA extracts. Wastewater samples were not found to have inhibition, but river water samples had inhibition at high iron concentrations and T3 qPCR was performed with 1:10 diluted samples and 1:100 diluted samples for 10 and 25 mg Fe L$^{-1}$ concentrate samples. All HS2 qPCR measurements for seawater samples were diluted 1:10. Each sample was measured in duplicate and the geometric mean was reported.

## Phage and 16S rRNA ddPCR assays

Singlet ddPCR reactions were performed with the QX200 AutoDG Droplet Digital PCR System (Bio-Rad Laboratories, Inc., Hercules, CA) with at least two ddH$_2$O negative controls and one positive control DNA extract from virus stocks per 96-well plate. Samples were multiplexed with two targets per reaction. Specific primer, probe, and annealing temperatures are provided in Table S10. 22 μL reactions contained 11 μL of 2x ddPCR$^{TM}$ Supermix for Probes (No dUTP) (Bio-Rad Laboratories, Inc., Cat. No. 1863023), 0.4 μM of all probes and primers, and 3 μL of DNA template. Droplets were generated to a 20 μL reaction volume using the automated droplet generation oil for probes (Bio-Rad Laboratories, Inc., Cat. No. 1864110) and the plate was sealed. PCR was performed on the C1000 Touch$^{TM}$ Thermal Cycler (Bio-Rad Laboratories, Inc., Hercules, CA) within 15 min of droplet generation with reaction conditions provided in Table S11. Plates were run on the droplet reader within 1 h of PCR completion with the exception of one plate that was stored at 4 °C for 60 h due to an error with the droplet reader. Thresholds were set for each
ddPCR reaction to determine the absolute abundance of 16S rRNA amplicons and phage amplicons using a previously defined method from Lievens et al. that categorizes droplets as positive, negative, or rain based on kernel density estimates for each reaction (*Lievens et al., 2016*). Reactions were rerun if there were more than two fluorescence populations, more than 2.5% of droplets were classified as rain, or less than 30% compartmentalization. Background signals were present in all 16S rRNA ddH$_2$O negative controls ($n = 22$, geometric mean = 263 gc $\mu$L$^{-1}$, 99% CI [233–305] gc $\mu$L$^{-1}$), as observed in previous studies (*Rehbinder et al., 2018*; *Sze et al., 2014*; *Dickson et al., 2018*). The 16S rRNA negative controls were significantly less than sample 16S rRNA measurements ($p$-value <0.000001) with 16S rRNA concentrations greater than the upper limit of the 99% confidence interval deemed acceptable (i.e., limit of quantification = 305 gc $\mu$L$^{-1}$) (*Huggett, 2020*). The mean background signal concentration of each 16S rRNA ddPCR run was subtracted from 16S rRNA sample measurements for the respective run to correct for 16S rRNA background signal. Alternatively, negative controls for the virus assays rarely resulted in target detection (T3: $n = 3$, max = 6.2 gc $\mu$L$^{-1}$; T4: $n = 2$, max = 14.8 gc $\mu$L$^{-1}$; PhiX174: $n = 0$; HS2: $n = 1$, max = 2.8 gc $\mu$L$^{-1}$; HM1: $n = 1$, max = 13.3 gc $\mu$L$^{-1}$; ICBM5: $n = 1$, max = 2.7 gc $\mu$L$^{-1}$). Given that viruses were spiked into samples at $10^4$ gc $\mu$L prior to concentrating for spike-and-recovery experiments, the rare virus detections in negative controls was deemed negligible.

## DNA concentration and quality assessment

After the complete ultrafiltration and purification method and iron chloride flocculation and purification method, DNA concentration and fragmentation were assessed. The dsDNA concentration was measured with Qubit$^{TM}$ dsDNA HS Assay (Invitrogen$^{TM}$, Cat. No. Q32851) with 1 or 2 $\mu$L of DNA template added to each 200 $\mu$L assay. The ssDNA concentration was determined by taking the difference between the measurement from Qubit$^{TM}$ ssDNA Assay (Invitrogen$^{TM}$, Cat. No. Q10212) and the dsDNA measurement. The ssDNA assay was performed with 1 $\mu$L of DNA template added to each 200 $\mu$L assay. DNA fragmentation for each matrix and method (triplicates pooled, 9 total samples) was assessed by Agilent TapeStation for DNA lengths up to 60,000 bp (Agilent, Cat. No. 5067-5365) according to manufacturer protocols. TapeStation processing was carried out in the Advanced Genomics Core at the University of Michigan.

## Statistical analysis

All statistical analysis and graphs were completed in Prism (version 8.4.3, GraphPad Software, LLC). Reported means are geometric with their respective 95% confidence intervals (CI) included (Table S11 and S12). Single phage spike recovery experiments were assessed with one-way ANOVA with Tukey's multiple comparison tests to generate $p$-values (Tables S14 and S15). Multiple phage spike recovery experiments were assessed with two-way ANOVA with Tukey's multiple comparison test to generate $p$-values (Tables S14 and S15). The outcomes from the two methods were compared with paired two-tailed t-tests with Holm-Sidak method to correct for multiple comparisons on each tested matrix for the final dsDNA and ssDNA concentrations, virus concentration factors, and 16S rRNA
**Table 3 Virus and 16S rRNA concentration factors after DNase treatment for both methods in each matrix.** The geometric mean and geometric standard deviation from the triplicate data is provided. The methods were compared with individual t-tests corrected for multiple comparisons with the Holm-Sidak method were performed for each matrix and phage spike.

| Matrix | Phage spike | Virus concentration factor | | | 16S rRNA concentration factor | | |
|---|---|---|---|---|---|---|---|
| | | Ultrafiltration | Flocculation | *p*-values | Ultrafiltration | Flocculation | *p*-values |
| Influent | T3 | 220 (120, 420) | 7.6 (4.7, 12) | 8.4E−3 (**) | 0.062 (0.016, 0.23) | 6.4E−3 (1.8E−3, 0.023) | 0.15 (ns) |
| Secondary Effluent | T3 | 440 (360, 540) | 6.7 (1.5, 30) | 6.8E−4 (***) | 6.3 (3.0, 13) | 0.15 (0.030, 0.76) | 0.19 (ns) |
| | T4 | 200 (71, 550) | NA | NA | | | |
| | PhiX174 | 45 (20, 100) | NA | NA | | | |
| River Water | T3 | 410 (280, 610) | 7.6 (1.9, 30) | 2.4E−3 (**) | 0.94 (0.48, 1.8) | 0.39 (0.031, 5.0) | 0.33 (ns) |
| Seawater | HS2 | 150 (50, 450) | 25 (22, 28) | 0.055 (ns) | 1.6 (0.50, 5.1) | 2.8 (0.85, 9.3) | 0.33 (ns) |
| | HM1 | 250 (75, 850) | 48 (40, 58) | 0.055 (ns) | | | |
| | ICBM5 | 110 (55, 220) | 3.7 (2.3, 5.9) | 9.3E−3 (**) | | | |

concentration factors (Table 3 and Table S16). Significance for all comparisons was any *p*-value less than 0.05 for all tests.

### Data availability

Jar test qPCR data and stepwise ultrafiltration and purification and iron chloride flocculation and purification ddPCR data is available in csv format in the "Viral Concentration and Purification Methods" Github repository (github.com/klangenf/Viral-Concentration-and-Purification-Methods).

## RESULTS

Two methods to concentrate and purify viruses were evaluated in four distinct matrices using virus spike-and-recovery tests. First, iron chloride concentrations were optimized for flocculation in each matrix using jar tests. Then, a two-step ultrafiltration method was compared to iron chloride flocculation stepwise through concentration and purification. The methods were evaluated based on viral concentration factors, viral recoveries, 16S rRNA concentration factors, virus to 16S rRNA enrichments, and final DNA quantity and quality. Finally, multiple DNA viruses were spiked into effluent and seawater samples to determine the extent to which the performance of each method was virus-specific.

### Optimization of iron chloride flocculation

The highest virus recoveries in the concentrate of influent, secondary effluent, and river water were achieved with 25 mg Fe L$^{-1}$, where T3 genomes were recovered at 74%, 72%, and 44%, respectively (Fig. 1B), signaling successful flocculation and capture of the viruses. The lowest viral recoveries in the filtrate of influent, effluent, and river water were achieved with 25 mg Fe L$^{-1}$ where T3 genomes were recovered at 6.2%, 7.1%, and 1.4%, respectively, indicating successful flocculation. The sum of the concentrate and filtrate recoveries, which should theoretically equal 100% (Fig. 1A), were 80%, 79%, and 46% of the spiked viral genomes for influent, effluent, and river water, respectively. For influent and secondary effluent, the viral genomes recovered in the filtrate decreased with increasing iron concentrations up to 25 mg Fe L$^{-1}$. We postulated that the iron chloride flocculation

performance would continue to improve with increased iron chloride concentrations. However, this would require more oxalic acid resuspension buffer to dissolve the formed flocs and the increased volume would be counterproductive to concentrating the viruses. Furthermore, solubility limits of the oxalic acid were reached when preparation of a 2x more concentrated resuspension buffer was attempted (*John, Poulos & Schirmer, 2015a*). Based on these limitations, we concluded that an iron chloride concentration of 25 mg Fe $L^{-1}$ was the best option for recovering viral DNA in the freshwater samples.

For seawater, best recoveries in the concentrate were observed with Fe concentrations of 5, 10, and 25 mg $L^{-1}$, where 69%, 52%, and 67% of the spiked HS2 genomes were recovered, respectively (Fig. 1B). In the filtrate, low recoveries of HS2 genomes were observed with Fe concentrations of 5, 10, and 25 mg $L^{-1}$, where 1.5%, 0.3%, and 0.1% of the spiked viruses were recovered, respectively. The filtrate and concentrate recoveries summed to 70%, 52%, and 67% with 5, 10, and 25 mg Fe $L^{-1}$, respectively. Given that iron chloride flocculation performs similarly at 5, 10, and 25 mg Fe $L^{-1}$ and 5 mg Fe $L^{-1}$ requires the smallest volume of resuspension buffer, 5 mg Fe $L^{-1}$ was chosen for recovering viral DNA from seawater. Notably, at 1 mg Fe $L^{-1}$, the current standard in seawater flocculation (*John et al., 2011*), HS2 genome recoveries were 33% in the concentrate and 75% in the filtrate, indicative of poorer flocculation than with the higher Fe concentrations tested here.

The sum of the filtrate and filter concentrate recoveries was often less than 100%, regardless of matrix. Control experiments confirmed that T3 and HS2 genomes did not degrade over the length of the iron chloride flocculation process in any matrix (Table S6). Thus, we suspect that the loss of viral genomes was due to inefficiencies in dissolving the flocs from the filters, which reduced recoveries in the concentrate.

### Evaluation of virus concentrating and recovery

Overall, the viruses were concentrated more following ultrafiltration and purification than with iron chloride flocculation and purification (Table 3). Specifically, the iron chloride flocculation and purification approach resulted in T3 and HS2 concentration factors of 7.6, 6.7, 7.6, and 25 for influent, effluent, river water, and seawater, respectively. The ultrafiltration and purification approach resulted in T3 and HS2 genome concentration factors of 220, 440, 410, and 150-fold for influent, effluent, and river water, and seawater, respectively. These virus concentration factors for the entire concentration and purification approaches were the result of two effects, the volume reduction (volume$_{final}$/volume$_{initial}$) and the virus recovery through all of the processes. During iron chloride flocculation, the volume of the freshwater samples was reduced from 500-mL to 12.5-mL and seawater was reduced from 20-L to 100-mL. The amount that the volume is reduced with iron chloride flocculation depends on the mass of iron added to the sample, so increasing the initial volume will not increase the relative reduction in sample volume. Alternatively, for the ultrafiltration and purification approach, the volumes were reduced approximately 500-fold for influent and 1,000-fold for effluent, river water, and seawater. The T3 and HS2 genome recoveries after iron chloride flocculation and purification were 25%, 21%, 25%, and 15% in influent, effluent, river water, and seawater, respectively. The final recovery of

T3 and HS2 genomes after ultrafiltration and purification was 47%, 42%, 43%, and 18% in influent, effluent, river water, and seawater, respectively.

Virus concentration factors increased following the iron chloride flocculation and ultrafiltration concentration steps, as expected. The magnitude of the virus concentration factor correlated with the reduction in volume for iron chloride flocculation, which can account for the high virus concentration factors in seawater compared to freshwater samples (Fig. 3A). Virus concentration factors increased twice during the ultrafiltration and purification process due to two ultrafiltration steps (Fig. 4A). In the freshwater matrices, the higher virus concentration factors in effluent and river water compared to influent correlates to a greater reduction in volume. Conversely, influent and seawater virus concentration factors were similar despite a greater volume reduction for seawater due to poorer recovery of HS2 in seawater, as compared to T3 in freshwater.

Despite demonstrated concentration of viruses, all of the concentration methods resulted in statistically significant viral losses (Figs. 3B and 4B). The ranges of recoveries were 63–80% following iron chloride flocculation, 57–82% following tangential ultrafiltration, and 35–86% following dead-end ultrafiltration. Recovery significantly decreased in effluent and seawater following iron chloride flocculation (Table S14). Recovery significantly decreased in effluent and seawater after both tangential ultrafiltration and dead-end ultrafiltration steps and in river water after only tangential ultrafiltration (Table S15).

The purification steps aimed to remove non-viral DNA, not concentrate viruses, so changes in the virus concentration factor through chloroform and DNase treatments depended solely on virus recovery. DNase treatment caused significant viral genome losses following iron chloride flocculation, but not following ultrafiltration concentration. Specifically, the range of recoveries following DNase treatments were 25–48% and 80–120% following iron chloride flocculation and ultrafiltration, respectively. DNase treatment during iron chloride flocculation and purification resulted in statistically significant decreases in viral genome recoveries in the influent, effluent, and seawater samples (Table S14). DNase treatment during ultrafiltration and purification did not result in a statistically significant reduction in viral genome recoveries in any of the sample types.

Based on the larger volume reductions and higher virus recoveries, we concluded that ultrafiltration and purification outperforms iron chloride flocculation and purification, particularly in freshwater matrices.

## Non-viral DNA removal performance

Filtration, chloroform, and DNase treatments are purification steps that aim to enrich the viral genomes relative to other organisms' genomes in order to focus sequencing effort on viral DNA. Overall, greater than 98% and 97% of 16S rRNA were removed from all matrices, following ultrafiltration and purification and iron chloride flocculation and purification, respectively (Fig. S3). Ultimately, selectively concentrating viruses compared to non-viral DNA is most important. In every case, viruses were concentrated by a greater factor than 16S rRNA (Table 3). This relationship was evaluated using the virus to 16S rRNA enrichment factor, whereby an enrichment factor greater than one indicated that viruses were concentrated more than 16S rRNA gene copies. Virus to 16S rRNA enrichment

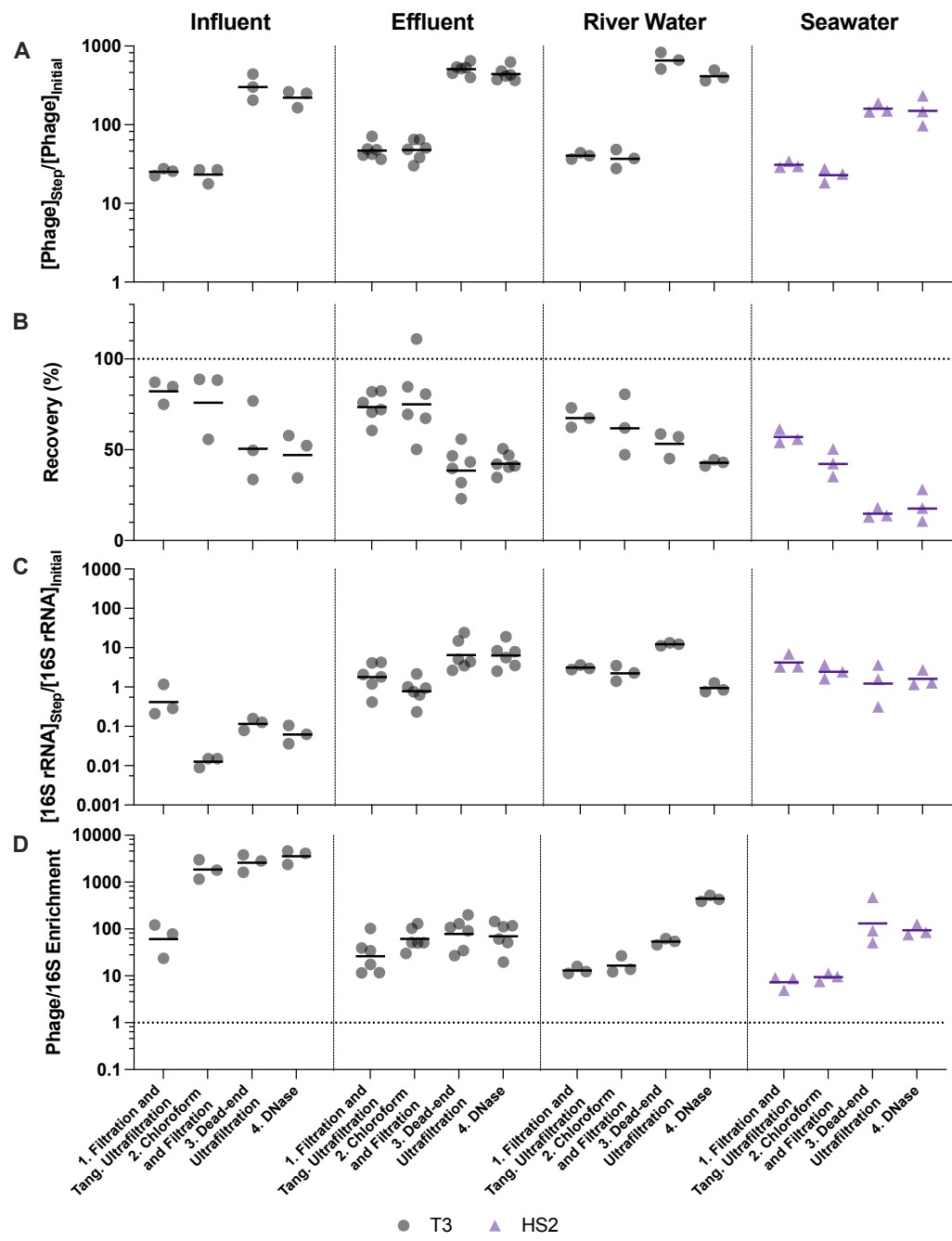

**Figure 4  Stepwise evaluation of ultrafiltration and purification with approximately 500-fold concentration by volume of influent and 1,000-fold concentration by volume of all other matrices.** (A) T3 or HS2 concentration factors on a log-scale at each step. Values greater than 1 indicated an increase in viral genomes. (B) T3 and HS2 recovery with each step of ultrafiltration showed losses of viruses throughout the process. Ideal recovery is indicated by the dotted line at 100%. (C) 16S rRNA concentration factor on a log-scale indicated non-viral DNA removal. Concentration factor values less than 1 indicated reduction of 16S rRNA gene copies. (D) T3 or HS2 to 16S rRNA enrichment with each step on a log-scale, as calculated by the virus concentration factor divided by the 16S rRNA concentration factor. Enrichments greater than 1 demonstrate viral genomes were concentrated more than 16S rRNA gene copies during the process. Individual measurements are points with a bar indicating the geometric mean of experimental replicates.

factors were 1000, 44, 19, and 8.8 after iron flocculation and purification and 3600, 70, 440, and 94 after ultrafiltration and purification for influent, effluent, river water, and seawater, respectively (Table S16). Virus to 16S rRNA enrichment factors for both methods were statistically greater than one, except after iron chloride flocculation and purification with river water (Table S14). Of the different sample types, the influent samples resulted in the lowest 16S rRNA concentration factors and highest virus to 16S rRNA enrichment factors with both methods, likely due to the high initial concentrations of 16S rRNA in influent relative to the other sample matrices (Fig. S3).

The purification steps are designed to enrich viral genomes. However, viral genomes in many sample types were not enriched by chloroform and DNase purification steps (Figs. 3D and 4D). During ultrafiltration and purification, enrichment of T3 relative to 16S rRNA was observed only in the influent samples following chloroform treatment and in the river water sample after DNase treatment. During iron chloride flocculation and purification, the only sample in which enrichment of T3 relative to 16S rRNA was observed was the seawater sample. Interestingly, the seawater samples that were concentrated with iron chloride flocculation and purification exhibited HS2 to 16S rRNA enrichment factors greater than one *only* following DNase treatment. This suggests that viral DNA was purified in this matrix by DNase or a combination of chloroform and DNase. With both concentration approaches, most of the 16S rRNA in freshwater was removed after the first step of pre-filtering (0.45-$\mu$m) and concentrating. This was likely because the prefiltration step removes a large fraction of the cells.

### Final DNA concentrations and fragmentation

Sequencing a sample requires a minimum quantity of DNA. Specifically, Illumina sequencing typically requires ~200 ng of DNA and Oxford Nanopore flow cells require ~1 $\mu$g of high molecular weight DNA. DNA yields ranged from 160 ng to 520 ng and 430 ng to 16 $\mu$g for iron chloride flocculation and ultrafiltration approaches, respectively (Fig. S4). These ranges are sufficient for generating amplification-free dsDNA and ssDNA virome libraries for Illumina sequencing. Following ultrafiltration and purification, secondary effluent yields were sufficient for Oxford Nanopore sequencing, but other matrices would require multiplexing with one or two additional samples per flow cell to be sufficient. An additional 10-fold virus DNA concentration would be necessary for amplification-free long read sequencing after applying the iron flocculation approach.

To evaluate whether final DNA extracts contained high molecular weight DNA, fragmentation was assessed with gel electrophoresis. Following the iron chloride flocculation method, DNA from freshwater samples resulted in faint streaks with darker regions below the 10 kb rung of the high range ladder and seawater samples had no visible DNA (Fig. S5). No T3 or HS2 genome bands were visible, suggesting the viral DNA had been sheared. All of the ultrafiltration and purification samples had clear high molecular weight DNA streaks between 10 and 50 kb with visible bands at the T3 or HS2 genome size (38 kb) indicating the genomes were not fragmented (Fig. S5). Together, the DNA concentration and fragmentation results suggest that the ultrafiltration and purification
method may be better suited for Nanopore sequencing than the iron chloride flocculation and purification method.

## Phage-specific recovery through concentration and purification

We expanded the number of phages tested under select experimental conditions to better understand the degree to which T3 and HS2 results were representative of other DNA viruses. As ultrafiltration and purification resulted in the greater viral concentration in freshwater, we applied this method to determine the concentration of two other phage types, dsDNA phage T4 and ssDNA phage PhiX174, in a freshwater secondary effluent sample (Table 2). As both the ultrafiltration and iron chloride flocculation methods provided equivalently effective viral concentration, we tested both methods with dsDNA phage HM1 and ssDNA phage ICBM5, in seawater.

Viral genome recoveries varied amongst the phages with both concentration methods. The iron chloride flocculation method applied to seawater resulted in genome recoveries of 15, 28, and 2.2% for HS2, HM1, and ICBM5, respectively. HM1 recovery was significantly greater than HS2 ($p$-value = 0.011) and ICBM5 ($p$-value = 1.9E-5) and HS2 recovery was significantly greater than ICBM5 ($p$-value = 0.021) (Fig. 5). HM1 and ICBM5 genome losses occurred at the same steps as the HS2 genome losses, namely following iron chloride flocculation and DNase treatment. The ultrafiltration and purification method applied to the secondary effluent resulted in genome recoveries of 42, 17, and 3.8% for T3, T4, and PhiX174, respectively, whereas T3 recovery was significantly greater than both T4 ($p$-value = 0.0058) and PhiX174 ($p$-value = 0.012). The ultrafiltration and purification method applied to the seawater resulted in genome recoveries of 18, 30, and 13% for HS2, HM1, and ICBM5, whereas HM1 recovery was significantly greater than HS2 ($p$-value = 0.083) and ICBM5 ($p$-value = 0.0098). The ssDNA phages, PhiX174 and ICBM5, had the lowest genome recovery of the spiked viruses after ultrafiltration and purification. As seen in the T3- and HS2-only experiments, the ultrafiltration steps resulted in the largest viral genome losses across the entire ultrafiltration and purification method. Following iron chloride flocculation, significant reductions in recovery after DNase treatment were common regardless of the virus or matrix (Table S14). Conversely, only ICBM5 recovery was significantly reduced by the DNase treatment following ultrafiltration ($p$-value = 0.0060).

## DISCUSSION

We compared two approaches for concentrating and purifying viruses, iron chloride flocculation and ultrafiltration, and applied them to four water matrices for the preparation of high-quality viral DNA extracts for metagenomics. Both concentration methods are widely used for virome studies (Brum et al., 2015; Beaulaurier et al., 2020; Gregory et al., 2019; Moon et al., 2020a; Moon et al., 2020b; Aguirre de Carcer et al., 2015; Roux et al., 2016a; Breitbart et al., 2002; Breitbart et al., 2003; Petrovich et al., 2019; Mohiuddin & Schellhorn, 2015; Skvortsov et al., 2016; Roux et al., 2012), but there has not been a systematic study evaluating stepwise performance and viral recovery through the concentration and purification processes across multiple matrices. As our end goal was the preparation of

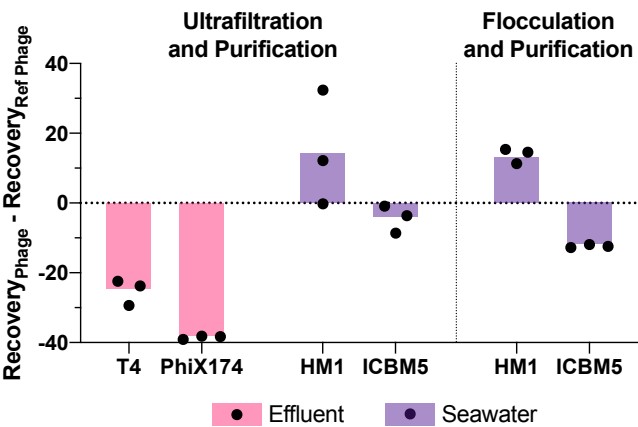

**Figure 5** **Variability in virus recovery depending on the phage was observed.** T3 in effluent and HS2 in seawater recoveries were used to evaluate the final genome recoveries of different phage types. The individual points are the difference between a final recovery individual measurement and the geometric mean final recovery of the reference phage in experimental replicates. In effluent, T4 and PhiX174 were compared to T3. In seawater, HM1 and ICBM5 were compared to HS2. The bars represent the mean of the individual points.

amplification-free sequencing libraries suitable for both short and long read sequencing, the ideal preparation method would generate sufficient mass of high molecular weight DNA for these purposes. As such, we evaluated iron chloride flocculation and ultrafiltration in terms of the resulting DNA concentration, purity, fragmentation, and the relative number of viral to cellular genome copies in the four different sample matrices.

## Five-fold more iron is needed to flocculate viruses from freshwater than seawater

Iron chloride flocculates viruses because it neutralizes the electrostatic repulsion layer of negatively charged particles. The behavior of this process is influenced by pH, the abundance of particles that vary in surface charge, and electrolyte strength, (*Jilbert et al., 2018*). Due to the dependence of flocculation efficiency on the relationship between sample matrix properties and flocculant concentration, we both expected and observed iron chloride flocculation performance to be strongly matrix-specific. The best iron chloride concentration for the flocculation and removal of viruses from freshwater matrices was 25 mg Fe $L^{-1}$. In contrast, seawater required less iron chloride, attaining equally high removal at concentrations at 5, 10, and 25 mg Fe $L^{-1}$. The concentrations required for maximal removal are higher than previously reported for both freshwater (*Chang et al., 1958*; *Manwaring, Chaudhuri & Engelbrecht, 1971*) and seawater (*John et al., 2011*) when a similar filtration-based approach was applied to capture flocs. This may be due to the fact that these previous studies only tested concentrations up to 10 and 1 mg Fe $L^{-1}$ for freshwater and seawater, respectively. Supporting the finding that less Fe was needed for flocculation in seawater, a recent estuary study demonstrated that increased iron flocculation occurred with increasing salinity (*Jilbert et al., 2018*). As more viral ecologists apply the commonly used iron chloride flocculation method first developed in

seawater (*John et al., 2011*) to freshwater samples, it is important to note that the best viral genome recovery in the freshwater samples was achieved at 25 mg Fe $L^{-1}$, 25-fold higher than the standard concentration used for ocean samples.

Iron chloride flocculation performance varied amongst the freshwater matrices (virus recoveries of 44–74%). The pH values of the different matrices were similar (range: 6.89–8.05, Table S1) and the T3 and HS2 have negative surface charges in this pH range (*Michen & Graule, 2010*; *Ghanem et al., 2016*). Consequently, the different flocculant concentrations necessary for optimized virus recoveries in the freshwater matrices were likely due to other water characteristics. For instance, viral genome recoveries in the freshwater jar tests decreased with increasing total solids content in the samples (Fig. 2, Table S1). Previous studies identified that flocculation performance decreased as solids content increased and additional iron was required for sufficient viral flocculation (*Chang et al., 1958*; *Manwaring, Chaudhuri & Engelbrecht, 1971*; *Leiknes, 2009*).

Our application of the jar test method, an elementary technique used to optimize flocculant concentrations in environmental engineering applications, emphasized the matrix-specific nature of iron chloride flocculation performance. We recommend these performance tests before applying iron chloride flocculation to recover viruses from novel matrices, especially when salinity and solids content are expected to differ from those tested here.

## Ultrafiltration and purification was best at concentrating viruses and preserving their genome integrity

Preparation of PCR-free sequencing libraries requires high masses of DNA (e.g., 200 ng for Illumina and 1 µg for Nanopore). Due to the low abundance of viral DNA relative to total community DNA in aquatic samples, the recovery of high viral DNA masses requires a concomitant significant reduction of water volumes. We concentrated seawater, river water, and secondary effluent approximately 1,000-fold and influent approximately 500-fold during ultrafiltration and purification to achieve sufficient DNA for sequencing. During the iron chloride flocculation process, only 40-fold concentration by volume of influent, effluent, and river water and 200-fold volume concentration for seawater was achieved. The iron chloride flocculation method is limited in its ability to concentrate viruses. Increasing the sample volume requires the addition of more iron chloride, which subsequently increases the required volume of resuspension buffer. Therefore, unlike with ultrafiltration, increasing sample volume does not ultimately increase the virus concentration factor.

Due to the ability of ultrafiltration to reduce sample volume more than iron chloride flocculation, we anticipated virus concentration factors to be larger after ultrafiltration and purification. In freshwater matrices, the ultrafiltration method concentrated T3 genomes 29, 66, and 54-fold more than iron chloride flocculation in influent, effluent, and river water, respectively. In seawater, the concentration factors obtained with the two methods were not different (6-fold more with ultrafiltration than with iron chloride flocculation). The limited ability for iron chloride flocculation to concentrate viruses in freshwater samples makes it a poorer choice for viral metagenomics applications, as the resultant DNA concentrations

are too low to generate sequencing libraries without applying additional steps. Two steps commonly applied to iron chloride flocculated samples include an additional concentration step, such as dead-end ultrafiltration (*Brum et al., 2015*; *Gregory et al., 2019*; *Roux et al., 2016a*; *Warwick-Dugdale et al., 2019*) and either PCR or enzymatic amplification of the genomic material (*Kim & Bae, 2011*; *Kim et al., 2008*; *Laver et al., 2016*; *Zhang et al., 2006*; *Woyke et al., 2009*). Our results indicate that, given its ability to generate sufficient viral DNA for amplification-free sequencing, ultrafiltration and purification is the more suitable method for quantitative virome studies, where biases in sequence representation must be minimized.

Long-read data can capture complete viral genomes with single reads (*Beaulaurier et al., 2020*), overcoming the challenges of assembling viral genomes from short-read data. Long-read sequencing, however, requires a large mass of high integrity, high molecular weight DNA. Our results suggest that the iron chloride flocculation concentration and purification method caused more DNA fragmentation than the ultrafiltration concentration and purification method. Two ocean virome studies have successfully applied long-read sequencing to obtain reads that were several kilobases long, with median lengths of 30 kb (*Beaulaurier et al., 2020*) and 4 kb (*Warwick-Dugdale et al., 2019*). *Beaulaurier et al. (2020)* successfully applied microfiltration followed by tangential ultrafiltration to generate high masses of high molecular weight DNA for amplification-free Nanopore libraries. Whereas, *Warwick-Dugdale et al. (2019)* used iron chloride flocculation followed by dead-end ultrafiltration. This study used 100 ng of input DNA and required PCR-adaptor ligation amplification, which reduced their read length potential to less than 8 kb. Our findings are consistent with these limited studies, namely that ultrafiltration concentration approaches produce greater viral concentration factors and higher quality DNA than iron chloride flocculation and are thus more suitable for long-read sequencing applications.

### Viral DNA is enriched more by ultrafiltration than iron chloride flocculation, while 16S rRNA is removed equally well during purification

Minimizing non-viral DNA contamination focuses sequencing effort on viral DNA and facilitates the capture of low abundance viruses. Non-viral DNA contamination was evaluated by the 16S rRNA gene concentration factor and the virus to 16S rRNA enrichment. 18S rRNA gene concentrations were not assessed in this study, as previous studies have demonstrated greater contamination of prokaryotic DNA in viromes as compared to eukaryotic DNA (*Hurwitz et al., 2013*; *Kleiner, Hooper & Duerkop, 2015*; *Moon et al., 2020b*). Both methods performed similarly at removing 16S rRNA gene copies, but the greater ability to concentrate viruses with the ultrafiltration steps resulted in a greater enrichment of viral genomes relative to 16S rRNA gene copies. The similar final 16S rRNA removals, 97–99% (Fig. S3), indicated we had reached a threshold for 16S rRNA removal. The remaining 16S rRNA gene copies may be encapsidated in gene transfer agents (*Lang, Zhaxybayeva & Beatty, 2012*) or otherwise protected from removal by the chloroform and DNase purification processes.
## Iron chloride flocculation disrupted inhibition of DNase reactions causing viral genome loss

DNase treatment was found to decrease virus recovery after iron chloride flocculation, but not after ultrafiltration. This suggests a mechanism whereby the iron chloride flocculation and purification method increased viral genome susceptibility to DNase enzymes. DNase activity requires calcium and magnesium ions; the DNase activity is therefore inhibited prior to genome extraction by adding EDTA and EGTA to chelate calcium and magnesium ions. The high level of $Fe^{3+}$ in these samples may have reduced the effectiveness of the EDTA at quenching the DNase activity prior to DNA extraction, thus leading to viral DNA degradation. Alternative viral purification methods to DNase or other approaches to DNase inhibition may improve viral retention during purification following iron chloride flocculation.

## Phage-specific recovery in both ultrafiltration and iron chloride flocculation is biased against ssDNA viruses

Significant phage-specific genome recoveries were observed with both methods. Regardless of the concentration method, genomes of the ssDNA phages, PhiX174 and ICBM5, were recovered less than those of the dsDNA phage genomes at the end of the concentration and purification processes. For all matrices concentrated with ultrafiltration, no single step in the method resulted in statistically significantly higher losses of ICBM5 or PhiX174, as compared to losses of the dsDNA viruses. However, by the end of the ultrafiltration method, the ssDNA virus recoveries were statistically lower than those of dsDNA viruses. In addition to having ssDNA genomes, PhiX174 and ICBM5 are smaller in diameter and have shorter genomes than the other viruses used in this study (Table 1). Previous studies have shown that smaller phages have poorer particle (*Ye et al., 2016*) and genome (*Kleiner, Hooper & Duerkop, 2015*) recoveries than larger dsDNA phages. Our observed differences in final recoveries demonstrated a source of bias that could impact downstream viral community representations in virome data. In the case of processing seawater, we observed greater variance in final virus recoveries with iron chloride flocculation and purification than with ultrafiltration and purification. Previously, *Hurwitz et al. (2013)* compared the concentration of seawater viruses with tangential ultrafiltration and iron chloride flocculation, and then evaluated the viromes produced by each protocol. Viromes prepared from iron chloride flocculation had more viral reads relative to non-viral and captured more rare viral reads than observed using tangential ultrafiltration. It was concluded that iron chloride flocculation introduced fewer virus-specific biases than tangential ultrafiltration, in contrast with our results. Given the different downstream purification methods applied (cesium chloride density gradient (*Hurwitz et al., 2013*) versus chloroform/filtration and DNase), direct comparisons of our results were not possible. To resolve this uncertainty, additional work with an expanded set of dsDNA and ssDNA viruses will be needed to conclusively evaluate the relative impact of iron chloride flocculation versus ultrafiltration on the diversity of recovered viruses.

## Limitations of phage spike-and-recovery experiments, selected matrices, and focus on DNA viruses

The spiked phages did not represent all known virus diversity. We selected the six tested phages to span a range of characteristics common to aquatic viruses, such as genome length, particle size, icosahedral capsid shapes and tails (Table 1). Although eukaryotic viruses were absent, phages are commonly accepted as surrogates for eukaryotic viruses in spike-and-recovery experiments (*Zhang et al., 2013*; *Rhodes et al., 2016*; *Pecson, Martin & Kohn, 2009*). Enveloped, double jelly roll capsid, and filamentous phages were absent from our spiked viruses, which has implications for the generalizability of our findings. Although enveloped viruses are considered a minor fraction of the known aquatic viromes (*Ye et al., 2016*; *Gundy, Gerba & Pepper, 2008*; *Casanova & Weaver, 2015*), inovirus filamentous phages were recently deemed more widespread and pervasive in the environment than previously thought (*Roux et al., 2019*). Enveloped, double jelly roll capsid, and filamentous viruses are known to lose infectivity when exposed to chloroform (*Kauffman et al., 2018*; *Petrenko et al., 1996*; *Taniguchi et al., 1984*), but this does not absolutely render their genomes susceptible to DNase enzymes that follow. Enveloped herpes virus genome recovery was not impacted by chloroform treatment (*Breitbart & Rohwer, 2005*; *Conceicao-Neto et al., 2015*), whereas the recoveries of enveloped coronavirus and mimivirus genomes, and those of enveloped reverse-transcribing viruses broadly, have been found to decrease following chloroform treatment (*Conceicao-Neto et al., 2015*; *Weynberg et al., 2014*). Although chloroform introduces biases, it facilitates the removal of 16S rRNA gene copies in subsequent DNase steps (SI section 4). Further investigation of the impact of chloroform treatment on viromes is needed to weigh the benefits (less cellular DNA, more low abundance viruses, lower limits of virus detection and quantification) versus drawbacks (possible biases in viral community representation) of this purification step.

All samples tested here fell within a narrow pH range (6.9–8.2), and some waters of interest may be outside of this range (e.g., acid mine drainage, alkaline lakes, treated drinking water). The pH of the water matrix impacts the overall charge of viruses and this may affect virus recoveries in the tested methods. Virus isoelectric points are typically less than seven (*Michen & Graule, 2010*; *Mayer et al., 2015*), although some have values as high as 8.4 (*Michen & Graule, 2010*). Viruses with isoelectric points greater than the pH range of our samples were not used in spike-and-recovery experiments. Since pH will change viral surface charges and the charge of the dominant iron species, applying iron chloride flocculation to matrices that have pH values deviating greatly from circumneutral may affect virus recoveries. An expanded set of viruses and water samples should be tested in the future to assess the impact of pH on virus recoveries.

This work focuses on methods that recover and purify DNA viruses for preparing viromes from seawater and freshwater samples. RNA viruses are also important members of viral communities, estimated to represent half of ocean viruses (*Steward et al., 2013*). Recent work has also expanded our knowledge of RNA phages in seawater (*Wolf et al., 2020*), though less work has been done on the prevalence and diversity of freshwater RNA viruses. Methods designed for effectively concentrating and purifying DNA viral genomes may not be as effective for RNA viruses. The necessary reverse transcription step

to form cDNA and the need to deplete host RNA, for example, add unique challenges in preparing unbiased RNA viral communities from sequencing (*Hong, Mantilla-Calderon & Wang, 2020*). The unknown relative abundance of RNA viruses in water samples further complicates RNA viral library preparation. Future work is therefore necessary to assess which methods work best for capturing RNA viral metagenomes, as well as the impact of initial relative abundance on the recovery of a given virus type through concentration and purification processes.

## CONCLUSIONS

We demonstrated the importance of assessing viral genome recovery and non-viral DNA removal prior to sequencing. Differences in aquatic matrices alter concentration and purification performances. Assuming a method performs adequately across matrices is inadvisable, as evidenced by the five-fold differences in best iron chloride concentrations for the freshwater and seawater samples. The ultrafiltration and purification method resulted in higher virus concentration factors and higher concentrations of high molecular weight DNA than iron chloride flocculation and purification for all tested matrices. We demonstrated that our ultrafiltration and purification protocol was superior to iron chloride flocculation and purification for influent, effluent, river water, and seawater samples. Given the demonstrated impact of solids content and salinity on the performance of these concentration and purification methods, we encourage future virome studies with matrices not tested here to assess virus concentration factors, recovery, and non-viral DNA removal with spike-and-recovery tests prior to sample preparation.

## ACKNOWLEDGEMENTS

We would like to thank Michael Mata for helping process influent and secondary effluent jar tests. We would like to thank the Duhaime and Wigginton Labs for contributing their technical and intellectual expertise for improvements of experimental design and manuscript preparation, as well as the Moraru Phage Lab (Institute for Chemistry and Biology of the Marine Environment, Oldenburg, Germany) for supplying phage ICBM5 and its host for spike-and-recovery studies.

### Funding

This project was funded by NSF PIRE Halting Environmental Antimicrobial Resistance Dissemination (HEARD) (Project No. 1545756) and the USDA (Project No. 2016-68003-24601). Kathryn Langenfeld was funded by an NSF GRFP (Fellow ID 2016216003), University of Michigan Jack A. Borchardt Fellowship, and University of Michigan Integrated Training in Microbial Systems (ITiMS) Fellowship that is funded by the Burroughs Wellcome Fund. The funders had no role in study design, data collection and analysis, decision to publish, or preparation of the manuscript.

## Grant Disclosures

The following grant information was disclosed by the authors:

NSF PIRE Halting Environmental Antimicrobial Resistance Dissemination (HEARD): 1545756.

USDA: 2016-68003-24601.

NSF GRFP: 2016216003.

Burroughs Wellcome Fund.

## Competing Interests

The authors declare there are no competing interests.

## Author Contributions

- Kathryn Langenfeld conceived and designed the experiments, performed the experiments, analyzed the data, prepared figures and/or tables, authored or reviewed drafts of the paper, and approved the final draft.
- Kaitlyn Chin and Ariel Roy performed the experiments, prepared figures and/or tables, and approved the final draft.
- Krista Wigginton and Melissa B. Duhaime conceived and designed the experiments, authored or reviewed drafts of the paper, and approved the final draft.

## Data Availability

Raw data is available at GitHub:

https://github.com/klangenf/Viral-Concentration-and-Purification-Methods.

## Supplemental Information

Supplemental information for this article can be found online at http://dx.doi.org/10.7717/peerj.11111#supplemental-information.

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
