# Peer review of "Comparison of ultrafiltration and iron chloride flocculation in the preparation of aquatic viromes from contrasting sample types"

_PeerJ, doi:10.7717/peerj.11111_

## Round 0.1 · original submission · Minor Revisions

We will be pleased to accept your manuscript for publication once a number of minor revisions are made to address the concerns of the reviewers.

Reviewer 1 ·

Basic reporting

The article is clear and well written, it has lots of side experiments, which enrich the experimental structure. Data included in figures and tables, as well as in supplementary material is really well presented. But, the introduction needs more detail on different ultrafiltration applications enhancing the references using rexeed ultrafilters and the background they got. Rexeed ultrafilters are dead-end ultrafiltration devices, that can be also modified to perform axial ultrafiltration (see Gallardo et al 2009). To my knowledge, Rexeed ultrafilters are not tangential flow ultrafilters, and the reference given by authors (22) does not use rexeed filters, use Fresenius filters which are different from rexeed.
Although relevant results are presented, authors should verify and correct this information in order not to confuse readers.

Experimental design

Research question is well defined and experiments rigorously performed. Although it is relevant to the audience that authors have included 6 different non-enveloped phages, they should point that the inclusion of an enveloped virus is lack on the study.
specific comments:
all p in p-values should be in italics
line 269: above instead of below?
line 387: delete "and" before river

Validity of the findings

no comment

Reviewer 2 ·

Basic reporting

A few citations are needed.
Line 56: According to the citation provided by the authors, viruses outnumber other microbes by up to two orders of magnitude, but in the following paragraph (lines 63 to 64) “viral DNA comprises a small fraction of total community DNA.” Include a literature citation for lines 63 to 64. Also, it is important to clarify how both ideas are true. Provide a description with an example that relates genome sizes of viruses to other microbes.

Line 541: Provide literature citation for “T3 and HS2 have negative surface charges in this pH range.”

Experimental design

The agarose gel electrophoresis experiment was challenging to interpret. Figure S5 did not contain information to determine the sizes of the marker DNA. The figure needs to be amended or better described in the text. Pulse field gel electrophoresis with appropriate ladders is the recommended method for environmental DNA virus genome quality analysis.

Validity of the findings

no comment

Additional comments

For authors
The authors compared two methods for purifying DNA viruses (ultrafiltration purification and iron chloride flocculation purification) present in wastewater (influent and secondary effluent), river water and seawater. They concluded ultrafiltration purification performs better than the iron chloride flocculation. This study provides much needed new information for preparing DNA viruses for metagenomic sequencing.

General comments/questions per section

Introduction
Line 56: According to the citation provided by the authors, viruses outnumber other microbes by up to two orders of magnitude, but in the following paragraph (lines 63 to 64) “viral DNA comprises a small fraction of total community DNA.” Include a literature citation for lines 63 to 64. Also, it is important to clarify how both ideas are true. Provide a description with an example that relates genome sizes of viruses to other microbes.

Materials and Methods
Lines 130 to 142: Check that the statements concerning virus spikes are consistent with Table 2. Add Table 2 for: “In freshwater matrices, Enterobacteria phage T3 (GenBank accession no. NC_003298, ATCC® BAA-1025-B1TM) was spiked into samples (Table 2).” The virus spikes for the various matrices are described except for wastewater influent. Add this missing description to this paragraph.

Line 289: Provide line equations and LOQs for the qPCR assays. How many non-template and negative extraction controls were included in the qPCR assays? How were false positives dealt with if they were observed?

Line 305: Did storing the droplets at 4C up to 60 hours cause any artifacts prior to running the plates? Such as those described in lines 308 to 310; i.e., rain or % compartmentalization?

Line 306: Provide more details for setting thresholds for classification of positive and negative droplets. Provide LOQ for 16S rRNA gene ddPCR. How many non-template and negative extraction controls were included in the ddPCR assays? How were false positives dealt with if they were observed?

Lines 318 to 323: The DNA virus spikes ranged in size from 5 to 168 Kb (see Table 1). Pulse field gel electrophoresis is a better approach for assessing the quality of spikes as well as endogenous DNA viruses. Explain why 0.3% AGE was used instead of PFGE for assessing quality of DNA.

Lines 470 479: Figure S5 did not contain information to determine the sizes of the marker DNA. This makes it difficult for the reader to interpret the data. The standard practice for gel electrophoresis is to indicate on the gel image the kilobase sizes for molecular standards. The figure needs to be amended or better described in the text. Some of the agarose wells appear to contain stainable DNA, while others didn’t. What is the size of this DNA? A related question, what is the resolution of 0.3% agarose? As stated above, PFGE with appropriate ladders is the recommended electrophoresis method for genome quality analysis. GeneRuler High Range DNA ladder has fragment sizes ranging between 10 and 50 Kb. Why was this ladder used for environmental DNA viruses known to be larger than 50Kb? Two of the virus spikes (Table 1) have genomes approximately two (HM1) or three (T4) times larger than 50Kb.

Line 541: Provide literature citation for “T3 and HS2 have negative surface charges in this pH range.”

Lines 584 to 604: 16S rRNA gene ddPCR assesses the removal of bacteria. Explain why eukaryotes (via 18S rRNA gene ddPCR) were not assessed too.

Reviewer 3 ·

Basic reporting

In this manuscript, Langenfeld et al compare two different virus concentration methods (tangential flow filtration and iron chloride flocculation) used prior to DNA extraction for viral metagenomic studies. They compare the efficiency of recovery in different samples, from wastewater treatment to seawater. To measure recovery, they spiked the samples with known concentrations of viral isolates and used droplet digital PCR to measure the recovery of these viruses at each step of the process.

The manuscript is well written and is easy to follow. The references cited are appropriate. The figures are clear.

Experimental design

No comment.

Validity of the findings

As the field of viral metagenomics keeps on expanding, this study is important to make sure that the highest quality of the viral assemblages is maintain during the concentration process.

Additional comments

While the spike and recovery approach is good to be able to monitor each step, it would have also been interesting to have total viral counts of before and after viral concentration, as well as after the DNAse step, which would better represent the impact of the methods on the viral assemblages. Omitting the remaining of the viruses lowers the impact of the publication.

It is unclear why they used T3 on freshwater samples and HS2 on seawater samples. While I understand that they did this because T3 was isolated from a freshwater host (E. coli) and HS2 a marine host (Pseudoalteromonas), it makes the comparisons difficult between the two environments as each phage/virus behaves differently.

Line 214: It is unclear why they decided to filter out the microbial host using a 0.45 µm filter since it is well known that many prokaryotes will go through these filters. While they did look at 16S rRNA gene contamination, when proposing benchmarked protocols, even though this is not what was not the step the study focused on, each step counts in the process and it seems that this one (filtration for removal of microbial hosts) was slightly overlooked.

Here, only one phage was added at a time. It would have been interesting to have all the phages added at the same time and measure the recovery of each. What works for a phage may not work for another one. Without redoing all the experiments, it could be worth expanding in the discussion.

---

## Round 0.2 · accepted · Accept

Thank-you for your thorough attention to the reviewer comments and your manuscript revision that addresses their concerns. We are now happy to accept your manuscript for publication.